# On the distance between two neural networks and the stability of learning

**Jeremy Bernstein**
Caltech
bernstein@caltech.edu

**Arash Vahdat**
NVIDIA
avahdat@nvidia.com

**Yisong Yue**
Caltech
yyue@caltech.edu

**Ming-Yu Liu**
NVIDIA
mingyul@nvidia.com

## Abstract

This paper relates *parameter distance* to *gradient breakdown* for a broad class of nonlinear compositional functions. The analysis leads to a new distance function called *deep relative trust* and a descent lemma for neural networks. Since the resulting learning rule seems to require little to no learning rate tuning, it may unlock a simpler workflow for training deeper and more complex neural networks. The Python code used in this paper is here: `https://github.com/jxbz/fromage`.

## 1 Introduction

Gradient descent is the workhorse of deep learning. To decrease a loss function $\mathcal{L}(W)$, this simple algorithm pushes the neural network parameters $W$ along the negative gradient of the loss. One motivation for this rule is that it minimises the local loss surface under a quadratic trust region [1]:

$$\underbrace{W - \eta \nabla \mathcal{L}(W)}_{\text{gradient descent}} = W + \arg\min_{\Delta W} \Big[ \underbrace{\mathcal{L}(W) + \nabla\mathcal{L}(W)^T \Delta W}_{\text{local loss surface}} + \underbrace{\tfrac{1}{2\eta}\|\Delta W\|_2^2}_{\text{quadratic penalty}} \Big]. \tag{1}$$

Since a quadratic trust region does not capture the compositional structure of a neural network, it is difficult to choose the learning rate $\eta$ in practice. Goodfellow et al. [2, p. 424] advise:

> If you have time to tune only one hyperparameter, tune the learning rate.

Practitioners usually resort to tuning $\eta$ over a logarithmic grid. If that fails, they may try tuning a different optimisation algorithm such as Adam [3]. But the cost of grid search scales exponentially in the number of interacting neural networks, and state of the art techniques involving just two neural networks are difficult to tune [4] and often unstable [5]. Eliminating the need for learning rate grid search could enable new applications involving multiple competitive and/or cooperative networks.

With that goal in mind, this paper replaces the quadratic penalty appearing in Equation 1 with a novel distance function tailored to the compositional structure of deep neural networks.

**Our contributions:**

1. By a direct perturbation analysis of the neural network function and Jacobian, we derive a distance on neural networks called *deep relative trust*.
2. We show how to combine deep relative trust with the fundamental theorem of calculus to derive a descent lemma tailored to neural networks. From the new descent lemma, we derive a learning algorithm that we call Frobenius matched gradient descent—or *Fromage*. Fromage's learning rate has a clear interpretation as controlling the *relative size* of updates.
3. While Fromage is similar to the LARS optimiser [6], we make the observation that Fromage works well across a range of standard neural network benchmarks—including generative adversarial networks and natural language transformers—*without learning rate tuning*.

## 2 Entendámonos...

*...so we understand each other.*

The goal of this section is to review a few basics of deep learning, including heuristics commonly used in algorithm design and areas where current optimisation theory falls short. We shall see that a good notion of distance between neural networks is still a subject of active research.

### Deep learning basics

Deep learning seeks to fit a *neural network* function $f(W; x)$ with parameters $W$ to a dataset of $N$ input-output pairs $\{x_i, y_i\}_{i=1}^N$. If we let $\mathcal{L}_i := \mathcal{L}(f_i, y_i)$ measure the discrepancy between prediction $f_i := f(W; x_i)$ and target $y_i$, then learning proceeds by gradient descent on the *loss*: $\sum_{i=1}^N \mathcal{L}_i$.

Though various neural network architectures exist, we shall focus our theoretical effort on the *multilayer perceptron*, which already contains the most striking features of general neural networks: matrices, nonlinearities, and layers.

**Definition 1** (Multilayer perceptron). A multilayer perceptron is a function $f : \mathbb{R}^{n_0} \to \mathbb{R}^{n_L}$ composed of $L$ layers. The $l$th layer is a linear map $W_l : \mathbb{R}^{n_{l-1}} \to \mathbb{R}^{n_l}$ followed by a nonlinearity $\varphi : \mathbb{R} \to \mathbb{R}$ that is applied elementwise. We may describe the network in two complementary ways:

$$f(x) := \underbrace{\varphi \circ W_L}_{\text{layer } L} \circ \underbrace{\varphi \circ W_{L-1}}_{\text{layer } L-1} \circ ... \circ \underbrace{\varphi \circ W_1}_{\text{layer } 1}(x); \qquad (L \text{ layer network})$$

$$h_l(x) := \varphi(W_l\, h_{l-1}(x)); \qquad h_0(x) := x. \qquad (\text{hidden layer recursion})$$

Since we wish to fit the network via gradient descent, we shall be interested in the gradient of the loss with respect to the $l$th parameter matrix. This may be decomposed via the chain rule. Schematically:

$$\nabla_{W_l} \mathcal{L} = \frac{\partial \mathcal{L}}{\partial f} \cdot \frac{\partial f}{\partial h_l} \cdot \varphi'(W_l\, h_{l-1}) \cdot h_{l-1}. \qquad (2)$$

The famous backpropagation algorithm [7] observes that the second term may be decomposed over the layers of the network. Following the notation of Pennington et al. [8]:

**Proposition 1** (Jacobian of the multilayer perceptron). *Consider a multilayer perceptron with $L$ layers. The layer-$l$-to-output Jacobian $J_l$ is given by:*

$$J_l := \frac{\partial f(x)}{\partial h_l} = \frac{\partial f}{\partial h_{L-1}} \cdot \frac{\partial h_{L-1}}{\partial h_{L-2}} \cdot ... \cdot \frac{\partial h_{l+1}}{\partial h_l}$$
$$= \Phi'_L W_L \cdot \Phi'_{L-1} W_{L-1} \cdot ... \cdot \Phi'_{l+1} W_{l+1},$$

*where $\Phi'_k := \mathrm{diag}\left[\varphi'\left(W_k\, h_{k-1}(x)\right)\right]$ denotes the derivative of the nonlinearity at the $k$th layer.*

A key observation is that the network function $f$ and Jacobian $\frac{\partial f}{\partial h_l}$ share a common mathematical structure—a deep, layered composition. We shall exploit this in our theory.

### Empirical deep learning

The formulation of gradient descent given in Equation 1 quickly encounters a problem known as the *vanishing and exploding gradient problem*, where the scale of updates becomes miscalibrated with the scale of parameters in different layers of the network. Common tricks to ameliorate the problem include careful choice of weight initialisation [9], dividing out the gradient scale [3], gradient clipping [10] and directly controlling the relative size of updates to each layer [6, 11]. Each of these techniques has been adopted in numerous deep learning applications.

Still, there is a cost to using heuristic techniques. For instance, techniques that rely on careful initialisation may break down by the end of training, leading to instabilities that are difficult to trace. These heuristics also lead to a proliferation of hyperparameters that are difficult to interpret and tune. We may wonder, *why does our optimisation theory not handle these matters for us?*

**Deep learning optimisation theory**

It is common in optimisation theory to assume that the loss has Lipschitz continuous gradients:

$$\|\nabla\mathcal{L}(W + \Delta W) - \nabla\mathcal{L}(W)\|_2 \leq \frac{1}{\eta}\|\Delta W\|_2.$$

By a standard argument involving the quadratic penalty mentioned in the introduction, this assumption leads to gradient descent [1]. This assumption is ubiquitous to the point that it is often just referred to as *smoothness* [12]. It is used in a number of works in the context of deep learning optimisation [12–15], distributed training [16], and generative adversarial networks [17].

We conduct an empirical study that shows that neural networks do not have Lipschitz continuous gradients in practice. We do this for a 16 layer multilayer perceptron (Figure 1), and find that the gradient grows roughly exponentially in the size of a perturbation. See [18] for more empirical investigation, and [19] for more discussion on this point.

Several classical optimisation frameworks go beyond the quadratic penalty $\|\Delta W\|_2^2$ of Equation 1:

1. Mirror descent [20] replaces $\|\Delta W\|_2^2$ by a *Bregman divergence* appropriate to the geometry of the problem. This framework was studied in relation to deep learning [21, 22], but it is still unclear whether there is a Bregman divergence that models the compositional structure of neural networks.

2. Natural gradient descent [23] replaces $\|\Delta W\|_2^2$ by $\Delta W^T F \Delta W$. The *Riemannian metric* $F \in \mathbb{R}^{d \times d}$ should capture the geometry of the $d$-dimensional function class. Unfortunately, even writing down a $d \times d$ matrix is intractable for modern neural networks. Whilst some works explore tractable approximations [24], natural gradient descent is still a quadratic model of trust—we find that trust is lost far more catastrophically in deep networks.

3. Steepest descent [25] replaces $\|\Delta W\|_2^2$ by an arbitrary distance function. Neyshabur et al. [26] explore this approach using a rescaling invariant norm on paths through the network. The downside to the freedom of steepest descent is that it does not operate from first principles—so the chosen distance function may not match up to the problem at hand.

Other work operates outside of these frameworks. Several papers study the effect of architectural decisions on signal propagation through the network [8, 27–30], though these works usually neglect functional distance and curvature of the loss surface. Pennington and Bahri [31] do study curvature, but rely on random matrix models to make progress.

**Generative adversarial networks**

Neural networks can learn to generate samples from complex distributions. Generative adversarial learning [32] trains a discriminator network $D$ to classify data as real or fake, and a generator network $G$ is trained to fool $D$. Competition drives learning in both networks. Letting $\mathcal{V}$ denote the success rate of the discriminator, the learning process is described as:

$$\min_G \max_D \mathcal{V}(G, D).$$

Defining the optimal discriminator for a given generator as $D^*(G) := \arg\max_D \mathcal{V}(G, D)$. Then generative adversarial learning reduces to a straightforward minimisation over the parameters of the generator:

$$\min_G \max_D \mathcal{V}(G, D) \equiv \min_G \mathcal{V}(G, D^*(G)).$$

In practice this is solved as an inner-loop, outer-loop optimisation procedure where $k$ steps of gradient descent are performed on the discriminator, followed by 1 step on the generator. For example, Miyato et al. [33] take $k = 5$ and Brock et al. [5] take $k = 2$.

For small $k$, this procedure is only well founded if the perturbation $\Delta G$ to the generator is small so as to induce a small perturbation in the optimal discriminator. In symbols, we hope that:

$$\|\Delta G\| \ll 1 \implies \|D^*(G + \Delta G) - D^*(G)\| \ll 1.$$

But what does $\|\Delta G\|$ mean? In what sense should it be small? We see that this is another area that could benefit from an appropriate notion of distance on neural networks.

# 3 The distance between networks

*Suppose that a teacher wishes to assess a student's learning. Traditionally, they will assign the student homework and track their progress. What if, instead, they could peer inside the student's head and observe change directly in the synapses—would that not be better for everyone?*

We would like to establish a meaningful notion of functional distance for neural networks. This should let us know how far we can safely perturb a network's weights without needing to tune a learning rate and measure the effect on the loss empirically.

The quadratic penalty of Equation 1 is akin to assuming a Euclidean structure on the parameter space, where the squared Euclidean length of a parameter perturbation determines how fast the loss function breaks down. The main pitfall of this assumption is that it does not reflect the deep compositional structure of a neural network. We propose a new trust concept called *deep relative trust*. In this section, we shall motivate deep relative trust—the formal definition shall come in Section 4.

Deep relative trust will involve a product over relative perturbations to each layer of the network. For a first glimpse of how this structure arises, consider a simple network that multiplies its input $x \in \mathbb{R}$ by two scalars $a, b \in \mathbb{R}$. That is $f(x) = a \cdot b \cdot x$. Also consider perturbed function $\widetilde{f}(x) = \widetilde{a} \cdot \widetilde{b} \cdot x$ where $\widetilde{a} := a + \Delta a$ and $\widetilde{b} := b + \Delta b$. Then the relative difference obeys:

$$\frac{|\widetilde{f}(x) - f(x)|}{|f(x)|} \leq \left(1 + \frac{|\Delta a|}{|a|}\right)\left(1 + \frac{|\Delta b|}{|b|}\right) - 1. \tag{3}$$

The simple derivation may be found at the beginning of Appendix B. The key point is that the relative change in the composition of two operators depends on the product of the relative change in each operator. The same structure extends to the relative distance between two neural networks—both in terms of function and gradient—as we will now demonstrate.

**Theorem 1.** *Let $f$ be a multilayer perceptron with nonlinearity $\varphi$ and $L$ weight matrices $\{W_l\}_{l=1}^L$. Let $\widetilde{f}$ be a second network with the same architecture but different weight matrices $\{\widetilde{W}_l\}_{l=1}^L$. For convenience, define layerwise perturbation matrices $\{\Delta W_l := \widetilde{W}_l - W_l\}_{l=1}^L$.*

*Further suppose that the following two conditions hold:*

1. *Transmission. There exist $\alpha, \beta \geq 0$ such that $\forall x, y$:*

$$\alpha \cdot \|x\|_2 \leq \|\varphi(x)\|_2 \leq \beta \cdot \|x\|_2;$$
$$\alpha \cdot \|x - y\|_2 \leq \|\varphi(x) - \varphi(y)\|_2 \leq \beta \cdot \|x - y\|_2.$$

2. *Conditioning. All matrices $\{W_l\}_{l=1}^L$, $\{\widetilde{W}_l\}_{l=1}^L$ and perturbations $\{\Delta W_l\}_{l=1}^L$ have condition number (ratio of largest to smallest singular value) no larger than $\kappa$.*

*Then a) for all non-zero inputs $x \in \mathbb{R}^{n_0}$, the relative functional difference obeys:*

$$\frac{\|\widetilde{f}(x) - f(x)\|_2}{\|f(x)\|_2} \leq \left(\frac{\beta}{\alpha}\kappa^2\right)^L \left[\prod_{k=1}^L \left(1 + \frac{\|\Delta W_k\|_F}{\|W_k\|_F}\right) - 1\right].$$

*And b) the layer-$l$-to-output Jacobian satisfies:*

$$\frac{\left\|\frac{\partial \widetilde{f}}{\partial \widetilde{h}_l} - \frac{\partial f}{\partial h_l}\right\|_F}{\left\|\frac{\partial f}{\partial h_l}\right\|_F} \leq \left(\frac{\beta}{\alpha}\kappa^2\right)^{L-l} \left[\prod_{k=l+1}^L \frac{\beta}{\alpha}\left(1 + \frac{\|\Delta W_k\|_F}{\|W_k\|_F}\right) - 1\right].$$

The proof is given in the appendix. The closest existing result that we are aware of is [34, Lemma 2], which bounds functional distance but not Jacobian distance. Also, since [34, Lemma 2] only holds for small perturbations, it misses the product structure of Theorem 1. It is the product structure that encodes the interactions between perturbations to different layers.

In words, Theorem 1 says that the relative change of a multilayer perceptron in terms of both function and gradient is controlled by a product over relative perturbations to each layer. Bounding the relative

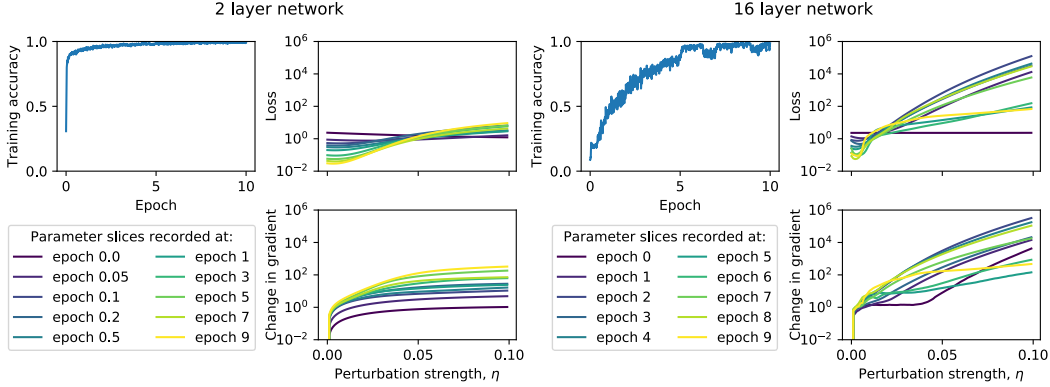

Figure 1: Using Fromage, we train a 2-layer (left) and 16-layer (right) perceptron to classify the MNIST dataset. With the network frozen at ten different training checkpoints, we first compute the gradient of the $l$th layer $g_l$ using the full data batch. We then record the loss and full batch gradient $\widetilde{g}_l$ after perturbing all weight matrices $W_l$ ($l = 1, ..., L$) to $W_l - \eta \cdot g_l \frac{\|W_l\|_F}{\|g_l\|_F}$ for various perturbation strengths $\eta$. We plot the classification loss and the relative change in gradient of the input layer $\|\widetilde{g}_1 - g_1\|_F / \|g_1\|_F$ along these parameter slices, all on a log scale. We find that the loss and relative change in gradient grow quasi-exponentially when the perceptron is deep, suggesting that Euclidean trust is violated. As such, these results are more consistent with our notion of deep relative trust.

change in function $f$ in terms of the relative change in parameters $W$ is reminiscent of a concept from numerical analysis known as the *relative condition number*. The relative condition number of a numerical technique measures the sensitivity of the technique to input perturbations. This suggests that we may think of Theorem 1 as establishing the relative condition number of a neural network with respect to parameter perturbations.

The two most striking consequences of Theorem 1 are that for a multilayer perceptron:

1. The trust region is not quadratic, but rather quasi-exponential at large depth $L$.

2. The trust region depends on the relative strength of perturbations to each layer.

To validate these theoretical consequences, Figure 1 displays the results of an empirical study. For two multilayer perceptrons of depth 2 and 16, we measured the stability of the loss and gradient to perturbation at various stages of training. For the depth 16 network, we found that the loss and gradient did break down quasi-exponentially in the layerwise relative size of a parameter perturbation. For the depth 2 network, the breakdown was much milder.

We will now discuss the plausibility of the assumptions made in Theorem 1. The first assumption amounts to assuming that the nonlinearity must transmit a certain fraction of its input. This is satisfied, for example, by the "leaky relu" nonlinearity, where for $0 < a \leq 1$:

$$\mathrm{leaky\_relu}(x) := \begin{cases} x & \text{if } x \geq 0; \\ ax & \text{if } x < 0. \end{cases}$$

Many nonlinearities only transmit half of their input domain—for example, the relu nonlinearity:

$$\mathrm{relu}(x) := \max(0, x).$$

Although relu is technically not covered by Theorem 1, we may model the fact that it transmits half its input domain by setting $\alpha = \beta = \frac{1}{2}$.

The second assumption is that all weight matrices and perturbations are full rank. In general this assumption may be violated. But provided that a small amount of noise is present in the updates, then by smoothed analysis of the matrix condition number [35, 36] it may often hold in practice.

# 4 Descent under deep relative trust

In the last section we studied the relative functional difference and gradient difference between two neural networks. Whilst the relative breakdown in gradient is intuitively an important object for optimisation theory, to make this intuition rigorous we have derived the following lemma:

**Lemma 1.** *Consider a continuously differentiable function $\mathcal{L} : \mathbb{R}^n \to \mathbb{R}$ that maps $W \mapsto \mathcal{L}(W)$. Suppose that parameter vector $W$ decomposes into $L$ parameter groups: $W = (W_1, W_2, ..., W_L)$, and consider making a perturbation $\Delta W = (\Delta W_1, \Delta W_2, ..., \Delta W_L)$. Let $\theta_l$ measure the angle between $\Delta W_l$ and negative gradient $-g_l(W) := -\nabla_{W_l}\mathcal{L}(W)$. Then:*

$$\mathcal{L}(W + \Delta W) - \mathcal{L}(W) \le -\sum_{l=1}^{L} \|g_l(W)\|_F \|\Delta W_l\|_F \left[\cos\theta_l - \max_{t\in[0,1]} \frac{\|g_l(W + t\Delta W) - g_l(W)\|_F}{\|g_l(W)\|_F}\right].$$

In words, the inequality says that the change in a function due to an input perturbation is bounded by the product of three terms, summed over parameter groups. The first two terms measure the size of the gradient and the size of the perturbation. The third term is a tradeoff between how aligned the perturbation is with the gradient, and how fast the gradient breaks down. Informally, the result says: *to reduce a function, follow the negative gradient until it breaks down.*

Descent is guaranteed when the bracketed term in Lemma 1 is positive. That is, for $l = 1, ..., L$:

$$\max_{t\in[0,1]} \frac{\|g_l(W + t\Delta W) - g_l(W)\|_F}{\|g_l(W)\|_F} < \cos\theta_l. \tag{4}$$

Geometrically this condition requires that, for every parameter group, the maximum change in gradient along the perturbation be smaller than the projection of the gradient in that direction. The simplest strategy to meet this condition is to choose $\Delta W_l = -\eta\, g_l(W)$. The learning rate $\eta$ must be chosen small enough such that, for all parameter groups $l$, the lefthand side of (4) is smaller than $\cos\theta_l = 1$. This is what is done in practice when gradient descent is used to train neural networks.

To do better than blindly tuning the learning rate in gradient descent, we need a way to estimate the relative gradient breakdown that appears in (4). The gradient of the loss is decomposed in Equation 2—let us inspect that result. Three of the four terms involve either a network Jacobian or a subnetwork output, for which relative change is governed by Theorem 1. Since Theorem 1 governs relative change in three of the four terms in Equation 2, we propose extending it to cover the whole expression—we call this modelling assumption *deep relative trust*.

**Modelling assumption 1** (Deep relative trust). *Consider a neural network with $L$ layers and parameters $W = (W_1, W_2, ..., W_L)$. Consider parameter perturbation $\Delta W = (\Delta W_1, \Delta W_2, ..., \Delta W_L)$. Let $g_l$ denote the gradient of the network loss function $\mathcal{L}$ with respect to parameter matrix $W_l$. Then the gradient breakdown is bounded by:*

$$\frac{\|g_l(W + \Delta W) - g_l(W)\|_F}{\|g_l(W)\|_F} \le \prod_{k=1}^{L}\left(1 + \frac{\|\Delta W_k\|_F}{\|W_k\|_F}\right) - 1.$$

Deep relative trust applies the functional form that appears in Theorem 1 to the gradient of the loss function. As compared to Theorem 1, we have set $\alpha = \beta = \frac{1}{2}$ (a model of relu) and $\kappa = 1$ (a model of well-conditioned matrices). Another reason that deep relative trust is a *model* rather than a *theorem* is that we have neglected the $\partial\mathcal{L}/\partial f$ term in Equation 2 which depends on the choice of loss function.

Given that deep relative trust appears to penalise the relative size of perturbations to each layer, it is natural that our learning algorithm should account for this by ensuring that layerwise perturbations are bounded like $\|\Delta W_l\|_F/\|W_l\|_F \le \eta$ for some small $\eta > 0$. The following lemma formalises this idea. As far as we are aware, this is the first descent lemma tailored to the neural network structure.

**Lemma 2.** *Let $\mathcal{L}$ be a continuously differentiable loss function for a neural network of depth $L$ that obeys deep relative trust. Consider a perturbation $\Delta W = (\Delta W_1, \Delta W_2, ..., \Delta W_L)$ to the parameters $W = (W_1, W_2, ..., W_L)$ with layerwise bounded relative size, meaning that $\|\Delta W_l\|_F/\|W_l\|_F \le \eta$ for $l = 1, ..., L$. Let $\theta_l$ measure the angle between $\Delta W_l$ and $-g_l(W)$. The perturbation will decrease the loss function provided that for all $l = 1, ..., L$:*

$$\eta < (1 + \cos\theta_l)^{\frac{1}{L}} - 1.$$

**Algorithm 1** *Fromage*—a good default $\eta = 0.01$.

---

**Input:** learning rate $\eta$ and matrices $\{W_l\}_{l=1}^{L}$
**repeat**
    collect gradients $\{g_l\}_{l=1}^{L}$
    **for** layer $l = 1$ **to** $L$ **do**
        $W_l \leftarrow \frac{1}{\sqrt{1+\eta^2}}\left[W_l - \eta \cdot \frac{\|W_l\|_F}{\|g_l\|_F} \cdot g_l\right]$
    **end for**
**until** converged

---

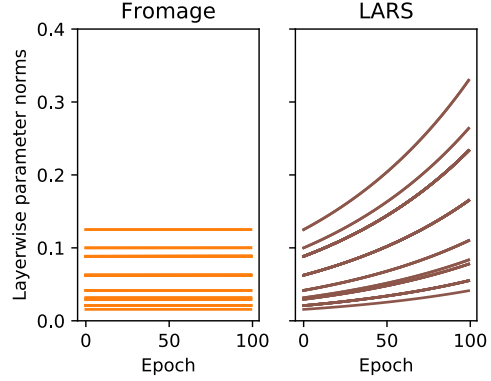

Figure 2: Left: the Fromage optimiser. Fromage differs to LARS [6] by the $1/\sqrt{1+\eta^2}$ prefactor. Right: without the prefactor, LARS suffers compounding growth in rescaling invariant layers that eventually leads to numerical overflow. This example is for a spectrally normalised cGAN [33, 37].

## 5   Frobenius matched gradient descent

In the previous section we established a descent lemma that takes into account the neural network structure. It is now time to apply that lemma to derive a learning algorithm.

The principle that we shall adopt is to make the largest perturbation that still guarantees descent by Lemma 2. To do this, we need to set $\cos \theta_l = 1$ whilst maintaining $\|\Delta W_l\|_F \leq \eta \|W_l\|_F$ for every layer $l = 1, ..., L$. This is achieved by the following update:

$$\Delta W_l = -\eta \cdot \frac{\|W_l\|_F}{\|g_l\|_F} \cdot g_l \qquad \text{for } l = 1, ..., L. \tag{5}$$

This learning rule is similar to LARS (layerwise adaptive rate scaling) proposed on empirical grounds by You et al. [6]. The authors demonstrated that LARS stabilises large batch network training.

Unfortunately, there is still an issue with this update rule that needs to be addressed. Neural network layers that involve *batch norm* [38] or *spectral norm* [33] are invariant to the rescaling map $W_l \rightarrow W_l + \alpha W_l$ and therefore the corresponding gradient $g_l$ must lie orthogonal to $W_l$. This means that the learning rule in Equation 2 will increase weight norms in these layers by a constant factor every iteration. To see this, we argue by Pythagoras' theorem combined with Equation 5:

$$\begin{aligned}\|W_l + \Delta W_l\|_F^2 \\ = \|W_l\|_F^2 + \|\Delta W_l\|_F^2 \\ = (1 + \eta^2) \|W_l\|_F^2.\end{aligned}$$

This is *compounding growth* meaning that it will lead to numerical overflow if left unchecked. We empirically validate this phenomenon in Figure 2. One way to solve this problem is to introduce and tune an additional weight decay hyperparameter, and this is the strategy adopted by LARS. In this work we are seeking to dispense with unnecessary hyperparameters, and therefore we propose explicitly correcting this instability via an appropriate prefactor. We call our algorithm Frobenius matched gradient descent—or *Fromage*. See Algorithm 1 above.

The attractive feature of Algorithm 1 is that there is only one hyperparameter and its meaning is clear. Neglecting the second order correction, we have that for every layer $l = 1, ..., L$, the algorithm's update satisfies:

$$\frac{\|\Delta W_l\|_F}{\|W_l\|_F} = \eta. \tag{6}$$

In words: the algorithm induces a relative change of $\eta$ in each layer of the neural network per iteration.

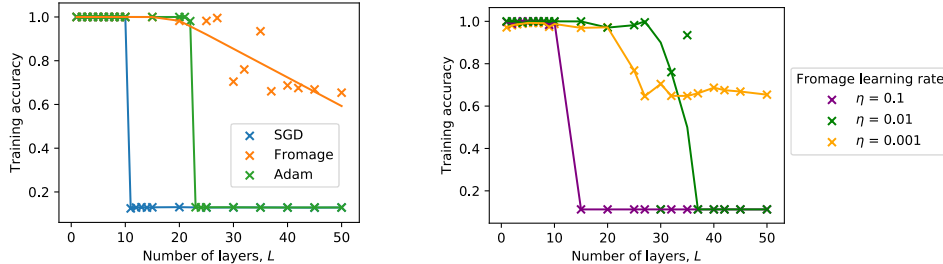

Figure 3: Training multilayer perceptrons at depths challenging for existing optimisers. We train multilayer perceptrons of depth $L$ on the MNIST dataset. At each depth, we plot the training accuracy after 100 epochs. Left: for each algorithm, we plot the best performing run over 3 learning rate settings found to be appropriate for that algorithm. We also plot trend lines to help guide the eye. Right: the Fromage results are presented for each learning rate setting. Since for deeper networks a smaller value of $\eta$ was needed in Fromage, these results provide partial support for Lemma 2.

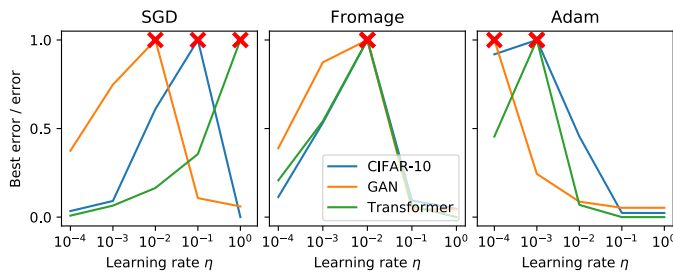

Figure 4: Learning rate tuning for standard benchmarks. For each learning rate setting $\eta$, we plot the error at the best tuned $\eta$ divided by the error for that $\eta$, so that a value of 1.0 corresponds to the best learning rate setting for that task. For Fromage, the setting of $\eta = 0.01$ was optimal across all tasks.

## 6   Empirical study

To test the main prediction of our theory—that the function and gradient of a deep network break down quasi-exponentially in the size of the perturbation—we directly study the behaviour of a multilayer perceptron trained on the MNIST dataset [39] under parameter perturbations. Perturbing along the gradient direction, we find that for a deep network the change in gradient and objective function indeed grow quasi-exponentially in the relative size of a parameter perturbation (Figure 1).

The theory also predicts that the geometry of trust becomes increasingly pathological as the network gets deeper, and Fromage is specifically designed to account for this. As such, it should be easier to train very deep networks with Fromage than by using other optimisers. Testing this, we find that off-the-shelf optimisers are unable to train multilayer perceptrons (without batch norm [38] or skip connections [40]) over 25 layers deep; Fromage was able to train up to at least depth 50 (Figure 3).

Next, we benchmark Fromage on four canonical deep learning tasks: classification of CIFAR-10 & ImageNet, generative adversarial network (GAN) training on CIFAR-10, and transformer training on Wikitext-2. In theory, Fromage should be easy to use because its one hyperparameter is meaningful for neural networks and is governed by a descent lemma (Lemma 2). The results are given in Table 1. Since this paper is about optimisation, all results are reported on the training set. Test set results and full experimental details are given in Appendix C. Fromage achieved lower training error than SGD on all four tasks (and by a substantial margin on three tasks). Fromage also outperformed Adam on three out of four tasks. Most importantly, Fromage used the same learning rate $\eta = 0.01$ across all tasks. In contrast, the learning rate in SGD and Adam needed to be carefully tuned, as shown in Figure 4. Note that for all algorithms, $\eta$ was decayed by 10 when the loss plateaued.

It should be noted that in preliminary runs of both the CIFAR-10 and Transformer experiments, Fromage would heavily overfit the training set. We were able to correct this behaviour for the results presented in Tables 1 and 2 by constraining the neural architecture. In particular, each layer's parameter norm was constrained to be no larger than its norm at initialisation.

Table 1: Training results. We quote loss for the classifiers, FID [4] for the GAN, and perplexity for the transformer—so lower is better. Test results and experimental details are given in Appendix C. For all algorithms, $\eta$ was decayed by 10 when the loss plateaued.

| Benchmark | SGD $\eta$ | Fromage $\eta$ | Adam $\eta$ | SGD | Fromage | Adam |
|---|---|---|---|---|---|---|
| CIFAR-10 | 0.1 | 0.01 | 0.001 | $(1.5 \pm 0.2) \times 10^{-4}$ | $\mathbf{(2.5 \pm 0.5) \times 10^{-5}}$ | $(6 \pm 3) \times 10^{-5}$ |
| ImageNet | 1.0 | 0.01 | 0.001 | $2.020 \pm 0.003$ | $\mathbf{2.001 \pm 0.001}$ | $2.02 \pm 0.01$ |
| GAN | 0.01 | 0.01 | 0.0001 | $34 \pm 2$ | $\mathbf{16 \pm 1}$ | $23.7 \pm 0.7$ |
| Transformer | 1.0 | 0.01 | 0.001 | $150.0 \pm 0.3$ | $66.1 \pm 0.1$ | $\mathbf{36.8 \pm 0.1}$ |

## 7 Discussion

The effort to alleviate learning rate tuning in deep learning is not new. For instance, Schaul et al. [41] tackled this problem in a paper titled "No More Pesky Learning Rates". Nonetheless—in practice—the need for learning rate tuning has persisted [2]. Then what sets our paper apart? The main contribution of our paper is its effort to explicitly model deep network gradients via *deep relative trust*, and to connect this model to the simple Fromage learning rule. In contrast, Schaul et al.'s work relies on estimating curvature information in an online fashion—thus incurring computational overhead while not providing clear insight on the *meaning* of the learning rate for neural networks.

To what extent may our paper eliminate pesky learning rates? Lemma 2 and the experiments in Figure 3 suggest that—for multilayer perceptrons—Fromage's optimal learning rate should depend on network depth. However, the results in Table 1 show that across a variety of more practically interesting benchmarks, Fromage's initial learning rate did not require tuning. It is important to point out that these practical benchmarks employ more involved network architectures than a simple multilayer perceptron. For instance, they use skip connections [40] which allow signals to skip layers. Future work could investigate how skip connections affect the deep relative trust model: an intuitive hypothesis is that they reduce some notion of *effective depth* of very deep networks.

In conclusion, we have derived a distance on deep neural networks called *deep relative trust*. We used this distance to derive a descent lemma for neural networks and a learning rule called *Fromage*. While the theory suggests a depth-dependent learning rate for multilayer perceptrons, we found that Fromage did not require learning rate tuning in our experiments on more modern network architectures.

## Broader impact

This paper aims to improve our foundational understanding of learning in neural networks. This could lead to many unforeseen consequences down the line. An immediate practical outcome of the work is a learning rule that seems to require less hyperparameter tuning than stochastic gradient descent. This may have the following consequences:

- Network training may become less time and energy intensive.
- It may become easier to train and deploy neural networks without human oversight.
- It may become possible to train more complex network architectures to solve new problems.

In short, this paper could make a powerful tool both easier to use and easier to abuse.

## Acknowledgements and disclosure of funding

The authors would like to thank Rumen Dangovski, Dillon Huff, Jeffrey Pennington, Florian Schaefer and Joel Tropp for useful conversations. They made heavy use of a codebase built by Jiahui Yu. They are grateful to Sivakumar Arayandi Thottakara, Jan Kautz, Sabu Nadarajan and Nithya Natesan for infrastructure support.

JB was supported by an NVIDIA fellowship, and this work was funded in part by NASA.

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
