[Supplementary Material]

## Appendix A    A descent lemma for neural networks

**Lemma 1.** *Consider a continuously differentiable function $\mathcal{L} : \mathbb{R}^n \to \mathbb{R}$ that maps $W \mapsto \mathcal{L}(W)$. Suppose that parameter vector $W$ decomposes into $L$ parameter groups: $W = (W_1, W_2, ..., W_L)$, and consider making a perturbation $\Delta W = (\Delta W_1, \Delta W_2, ..., \Delta W_L)$. Let $\theta_l$ measure the angle between $\Delta W_l$ and negative gradient $-g_l(W) := -\nabla_{W_l} \mathcal{L}(W)$. Then:*

$$\mathcal{L}(W + \Delta W) - \mathcal{L}(W) \le -\sum_{l=1}^{L} \|g_l(W)\|_F \|\Delta W_l\|_F \left[ \cos \theta_l - \max_{t \in [0,1]} \frac{\|g_l(W + t\Delta W) - g_l(W)\|_F}{\|g_l(W)\|_F} \right].$$

*Proof.* By the fundamental theorem of calculus,

$$\mathcal{L}(W + \Delta W) - \mathcal{L}(W) = \sum_{l=1}^{L} g_l(W)^T \Delta W_l + \int_0^1 \left[ g_l(W + t\Delta W) - g_l(W) \right]^T \Delta W_l \, \mathrm{d}t.$$

The result follows by replacing the first term on the righthand side by the cosine formula for the dot product, and bounding the second term via the integral estimation lemma. □

**Lemma 2.** *Let $\mathcal{L}$ be a continuously differentiable loss function for a neural network of depth $L$ that obeys deep relative trust. Consider a perturbation $\Delta W = (\Delta W_1, \Delta W_2, ..., \Delta W_L)$ to the parameters $W = (W_1, W_2, ..., W_L)$ with layerwise bounded relative size, meaning that $\|\Delta W_l\|_F / \|W_l\|_F \le \eta$ for $l = 1, ..., L$. Let $\theta_l$ measure the angle between $\Delta W_l$ and $-g_l(W)$. The perturbation will decrease the loss function provided that for all $l = 1, ..., L$:*

$$\eta < (1 + \cos \theta_l)^{\frac{1}{L}} - 1.$$

*Proof.* Using the gradient reliability estimate from deep relative trust, we obtain that:

$$\max_{t \in [0,1]} \frac{\|g_l(W + t\Delta W) - g_l(W)\|_2}{\|g_l(W)\|_2} \le \max_{t \in [0,1]} \prod_{k=1}^{L} \left( 1 + \frac{\|t\Delta W_k\|_F}{\|W_k\|_F} \right) - 1 \le \prod_{k=1}^{L} \left( 1 + \frac{\|\Delta W_k\|_F}{\|W_k\|_F} \right) - 1.$$

To guarantee descent, we require that the bracketed term in Lemma 1 is positive for all $l = 1, ..., L$. By the previous inequality, this will occur provided that for all $l = 1, ..., L$:

$$\prod_{k=1}^{L} \left( 1 + \frac{\|\Delta W_k\|_F}{\|W_k\|_F} \right) < 1 + \cos \theta_l.$$

Since $\|\Delta W_k\|_F / \|W_k\|_F \le \eta$ for $k = 1, ..., L$, this inequality will be satisfied provided that $(1 + \eta)^L < 1 + \cos \theta_l$. After a simple rearrangement, we are done. □

## Appendix B    A perturbation analysis of the multilayer perceptron

We begin by fleshing out the analysis of the two-layer scalar network (Equation 3), since this example already goes a long way to exposing the relevant mathematical structure.

Consider $f : \mathbb{R} \to \mathbb{R}$ defined by $f(x) = a \cdot b \cdot x$ for $a, b \in \mathbb{R}$. Also consider perturbed function $\widetilde{f}(x) = \widetilde{a} \cdot \widetilde{b} \cdot x$ where $\widetilde{a} := a + \Delta a$ and $\widetilde{b} := b + \Delta b$. The relative difference obeys:

$$\frac{|\widetilde{f}(x) - f(x)|}{|f(x)|} = \left| \frac{\widetilde{a}\widetilde{b}x - abx}{abx} \right| = \left| \frac{(a + \Delta a)(b + \Delta b) - ab}{ab} \right| = \left| \left( 1 + \frac{\Delta a}{a} \right) \left( 1 + \frac{\Delta b}{b} \right) - 1 \right|$$

$$\leq \left( 1 + \frac{|\Delta a|}{|a|} \right) \left( 1 + \frac{|\Delta b|}{|b|} \right) - 1.$$

Our main theorem will generalise this argument to more involved cases:

**Theorem 1.** *Let $f$ be a multilayer perceptron with nonlinearity $\varphi$ and $L$ weight matrices $\{W_l\}_{l=1}^L$. Let $\widetilde{f}$ be a second network with the same architecture but different weight matrices $\{\widetilde{W}_l\}_{l=1}^L$. For convenience, define layerwise perturbation matrices $\{\Delta W_l := \widetilde{W}_l - W_l\}_{l=1}^L$.*

*Further suppose that the following two conditions hold:*

1. *Transmission. There exist $\alpha, \beta \geq 0$ such that $\forall x, y$:*
$$\alpha \cdot \|x\|_2 \leq \|\varphi(x)\|_2 \leq \beta \cdot \|x\|_2;$$
$$\alpha \cdot \|x - y\|_2 \leq \|\varphi(x) - \varphi(y)\|_2 \leq \beta \cdot \|x - y\|_2.$$

2. *Conditioning. All matrices $\{W_l\}_{l=1}^L$, $\{\widetilde{W}_l\}_{l=1}^L$ and perturbations $\{\Delta W_l\}_{l=1}^L$ have condition number (ratio of largest to smallest singular value) no larger than $\kappa$.*

*Then a) for all non-zero inputs $x \in \mathbb{R}^{n_0}$, the relative functional difference obeys:*

$$\frac{\|\widetilde{f}(x) - f(x)\|_2}{\|f(x)\|_2} \leq \left( \frac{\beta}{\alpha} \kappa^2 \right)^L \left[ \prod_{k=1}^L \left( 1 + \frac{\|\Delta W_k\|_F}{\|W_k\|_F} \right) - 1 \right].$$

*And b) the layer-$l$-to-output Jacobian satisfies:*

$$\frac{\left\| \frac{\partial \widetilde{f}}{\partial h_l} - \frac{\partial f}{\partial h_l} \right\|_F}{\left\| \frac{\partial f}{\partial h_l} \right\|_F} \leq \left( \frac{\beta}{\alpha} \kappa^2 \right)^{L-l} \left[ \prod_{k=l+1}^L \frac{\beta}{\alpha} \left( 1 + \frac{\|\Delta W_k\|_F}{\|W_k\|_F} \right) - 1 \right].$$

Though we shall prove the two parts of this theorem separately, the following lemma shall help in both cases.

**Lemma 3** (Relative matrix-matrix conditioning). *Consider any two matrices $\widetilde{M}, M \in \mathbb{R}^{n \times m}$ with condition number bounded by $\kappa$. Then for any matrix $X$ and any non-zero matrix $Y$:*

$$\frac{\|\widetilde{M}X\|_F}{\|MY\|_F} \leq \kappa^2 \frac{\|\widetilde{M}\|_F \|X\|_F}{\|M\|_F \|Y\|_F}.$$

*Proof.* Denote the singular values of $M$ by $\sigma_1 \geq \sigma_2 \geq ... \geq \sigma_{\overline{n}}$ and the singular values of $\widetilde{M}$ by $\widetilde{\sigma}_1 \geq \widetilde{\sigma}_2 \geq ... \geq \widetilde{\sigma}_{\overline{n}}$. Observe that by letting $\widetilde{M}$ and $M$ act on the columns of $X$ and $Y$ (denoted $x_i$ and $y_j$ respectively) we have:

$$\frac{\|\widetilde{M}X\|_F^2}{\|MY\|_F^2} = \frac{\sum_i \|\widetilde{M}x_i\|_2^2}{\sum_j \|My_j\|_2^2} \leq \frac{\widetilde{\sigma}_1^2 \sum_i \|x_i\|_2^2}{\sigma_{\overline{n}}^2 \sum_j \|y_j\|_2^2} = \frac{\widetilde{\sigma}_1^2 \|X\|_F^2}{\sigma_{\overline{n}}^2 \|Y\|_F^2}.$$

Since it holds that $\sigma_1/\sigma_{\overline{n}} \leq \kappa$ and $\widetilde{\sigma}_1/\widetilde{\sigma}_{\overline{n}} \leq \kappa$, we obtain the following inequalities:

$$\|\widetilde{M}\|_F^2 = \sum_{i=1}^{\overline{n}} \widetilde{\sigma}_i^2 \geq \overline{n}\widetilde{\sigma}_{\overline{n}}^2 \geq \overline{n}\frac{\widetilde{\sigma}_1^2}{\kappa^2}.$$

$$\|M\|_F^2 = \sum_{i=1}^{\overline{n}} \sigma_i^2 \leq \overline{n}\sigma_1^2 \leq \overline{n}\kappa^2\sigma_{\overline{n}}^2.$$

To complete the proof, we substitute these two results into the first inequality to yield:

$$\frac{\|\widetilde{M}X\|_F}{\|MY\|_F} \le \frac{\widetilde{\sigma}_1\|X\|_F}{\sigma_{\overline{n}}\|Y\|_F} \le \kappa^2 \frac{\|\widetilde{M}\|_F\|X\|_F}{\|M\|_F\|Y\|_F}.$$

$\square$

With this tool in hand, let us proceed to prove part a) of Theorem 1.

*Proof of Theorem 1 part a).* To make an inductive argument, we shall assume that the result holds for a network with $L-1$ layers. Extending to depth $L$, we have:

$$\frac{\|\widetilde{f}(x) - f(x)\|_2}{\|f(x)\|_2} = \frac{\|(\varphi \circ \widetilde{W}_L) \circ \widetilde{h}_{L-1}(x) - (\varphi \circ W_L) \circ h_{L-1}(x)\|_2}{\|(\varphi \circ W_L) \circ h_{L-1}(x)\|_2}$$

$$\le \frac{\beta}{\alpha} \frac{\|\widetilde{W}_L \widetilde{h}_{L-1}(x) - W_L h_{L-1}(x)\|_2}{\|W_L h_{L-1}(x)\|_2} \qquad \text{(assumption on } \varphi)$$

$$= \frac{\beta}{\alpha} \frac{\|\Delta W_L \widetilde{h}_{L-1}(x) + W_L(\widetilde{h}_{L-1}(x) - h_{L-1}(x))\|_2}{\|W_L h_{L-1}(x)\|_2}$$

$$\le \frac{\beta}{\alpha} \frac{\|\Delta W_L \widetilde{h}_{L-1}(x)\|_2 + \|W_L(\widetilde{h}_{L-1}(x) - h_{L-1}(x))\|_2}{\|W_L h_{L-1}(x)\|_2} \qquad \text{(triangle inequality)}$$

$$\le \frac{\beta}{\alpha}\kappa^2 \left[ \frac{\|\Delta W_L\|_F}{\|W_L\|_F} \frac{\|\widetilde{h}_{L-1}(x)\|_2}{\|h_{L-1}(x)\|_2} + \frac{\|\widetilde{h}_{L-1}(x) - h_{L-1}(x)\|_2}{\|h_{L-1}(x)\|_2} \right]. \qquad \text{(Lemma 3)}$$

Whilst the second term may be bounded by the inductive hypothesis, we shall now show that the first term obeys:

$$\frac{\|\widetilde{h}_{L-1}(x)\|_2}{\|h_{L-1}(x)\|_2} \le \left(\frac{\beta}{\alpha}\kappa^2\right)^{L-1} \prod_{k=1}^{L-1} \left(1 + \frac{\|\Delta W_k\|_F}{\|W_k\|_F}\right).$$

We argue as follows:

$$\frac{\|\widetilde{h}_{L-1}(x)\|_2}{\|h_{L-1}(x)\|_2} = \frac{\|\varphi(\widetilde{W}_{L-1}\widetilde{h}_{L-2}(x))\|_2}{\|\varphi(W_{L-1}h_{L-2}(x))\|_2}$$

$$\le \frac{\beta}{\alpha} \frac{\|\widetilde{W}_{L-1}\widetilde{h}_{L-2}(x)\|_2}{\|W_{L-1}h_{L-2}(x)\|_2} \qquad \text{(assumption on } \varphi)$$

$$\le \frac{\beta}{\alpha}\kappa^2 \frac{\|\widetilde{W}_{L-1}\|_F}{\|W_{L-1}\|_F} \frac{\|\widetilde{h}_{L-2}(x)\|_2}{\|h_{L-2}(x)\|_2} \qquad \text{(Lemma 3)}$$

$$\le \frac{\beta}{\alpha}\kappa^2 \frac{\|W_{L-1}\|_F + \|\Delta W_{L-1}\|_F}{\|W_{L-1}\|_F} \frac{\|\widetilde{h}_{L-2}(x)\|_2}{\|h_{L-2}(x)\|_2} \qquad \text{(triangle inequality)}$$

$$= \frac{\beta}{\alpha}\kappa^2 \left(1 + \frac{\|\Delta W_{L-1}\|_F}{\|W_{L-1}\|_F}\right) \frac{\|\widetilde{h}_{L-2}(x)\|_2}{\|h_{L-2}(x)\|_2}.$$

The statement follows from an obvious induction on depth. Substituting this result and the inductive hypothesis back into the bound on $\|\widetilde{f}(x) - f(x)\|_2/\|f(x)\|_2$, we obtain:

$$\frac{\|\widetilde{f}(x) - f(x)\|_2}{\|f(x)\|_2} \le \left(\frac{\beta}{\alpha}\kappa^2\right)^L \left[ \frac{\|\Delta W_L\|_F}{\|W_L\|_F} \prod_{k=1}^{L-1}\left(1 + \frac{\|\Delta W_k\|_F}{\|W_k\|_F}\right) + \prod_{k=1}^{L-1}\left(1 + \frac{\|\Delta W_k\|_F}{\|W_k\|_F}\right) - 1 \right]$$

$$= \left(\frac{\beta}{\alpha}\kappa^2\right)^L \left[ \prod_{k=1}^{L}\left(1 + \frac{\|\Delta W_k\|_F}{\|W_k\|_F}\right) - 1 \right].$$

$\square$

Let us now proceed to the second part of Theorem 1.

*Proof of Theorem 1 part b).* By Proposition 1, the layer-$l$-to-output Jacobian $J_l$ satisfies:

$$J_l := \frac{\partial f(x)}{\partial h_l} = \Phi'_L W_L \cdot \Phi'_{L-1} W_{L-1} \cdot \ldots \cdot \Phi'_{l+1} W_{l+1}$$

where $\Phi'_k := \text{diag}\left[\varphi'\left(W_k h_{k-1}(x)\right)\right]$. Denote the perturbed version $\widetilde{J}_l$ by the product:

$$\widetilde{J}_l := \frac{\partial \widetilde{f}(x)}{\partial h_l} = (\Phi'_L + \Delta\Phi'_L)(W_L + \Delta W_L) \cdot \ldots \cdot (\Phi'_{l+1} + \Delta\Phi'_{l+1})(W_{l+1} + \Delta W_{l+1}).$$

For the purpose of an inductive argument, let us define the tail $T_l$ of the Jacobian to satisfy:

$$J_l = \Phi'_L W_L T_l$$

and similarly for the perturbed version $\widetilde{T}_l$. The inductive hypothesis then becomes:

$$\frac{\|\widetilde{T}_l - T_l\|_F}{\|T_l\|_F} \leq \left(\frac{\beta}{\alpha}\kappa^2\right)^{L-l-1}\left[\prod_{k=l+1}^{L-1}\frac{\beta}{\alpha}\left(1 + \frac{\|\Delta W_k\|_F}{\|W_k\|_F}\right) - 1\right].$$

We need to extend this to $J_l$. First note that by taking limits of the condition on the nonlinearity, we obtain that $0 \leq \alpha \leq \varphi'(x) \leq \beta$ for all $x$. This implies that for all layers $l$ the entries of the diagonal matrix $\Phi'_l$ lie between $\alpha$ and $\beta$ and the maximum entry of the diagonal matrix $\Delta\Phi'_l$ is no larger than $\beta - \alpha$. We shall use this information along with the triangle inequality and Lemma 3 to obtain the following:

$$\frac{\|\widetilde{J}_l - J_l\|_F}{\|J_l\|_F} = \frac{\|(\Phi'_L + \Delta\Phi'_L)(W_L + \Delta W_L)\widetilde{T}_l - \Phi'_L W_L T_l\|_F}{\|\Phi'_L W_L T_l\|_F}$$

$$= \frac{\|\Delta\Phi'_L(W_L + \Delta W_L)\widetilde{T}_l + \Phi'_L[(W_L + \Delta W_L)\widetilde{T}_l - W_L T_l]\|_F}{\|\Phi'_L W_L T_l\|_F}$$

$$\leq \frac{\|\Delta\Phi'_L(W_L + \Delta W_L)\widetilde{T}_l\|_F + \|\Phi'_L[(W_L + \Delta W_L)\widetilde{T}_l - W_L T_l]\|_F}{\|\Phi'_L W_L T_l\|_F}$$

$$\leq \frac{(\beta - \alpha)\|(W_L + \Delta W_L)\widetilde{T}_l\|_F + \beta\|(W_L + \Delta W_L)\widetilde{T}_l - W_L T_l\|_F}{\alpha\|W_L T_l\|_F}$$

$$\leq \frac{(\beta - \alpha)\|(W_L + \Delta W_L)\widetilde{T}_l\|_F + \beta\|\Delta W_L \widetilde{T}_l\|_F + \beta\|W_L(\widetilde{T}_l - T_l)\|_F}{\alpha\|W_L T_l\|_F}$$

$$\leq \kappa^2\frac{(\beta - \alpha))\|W_L + \Delta W_L\|_F\|\widetilde{T}_l\|_F + \beta\|\Delta W_L\|_F\|\widetilde{T}_l\|_F + \beta\|W_L\|_F\|\widetilde{T}_l - T_l\|_F}{\alpha\|W_L\|_F\|T_l\|_F}$$

$$\leq \kappa^2\left[\frac{\beta - \alpha}{\alpha}\left(1 + \frac{\|\Delta W_L\|_F}{\|W_L\|_F}\right)\frac{\|\widetilde{T}_l\|_F}{\|T_l\|_F} + \frac{\beta}{\alpha}\frac{\|\Delta W_L\|_F}{\|W_L\|_F}\frac{\|\widetilde{T}_l\|_F}{\|T_l\|_F} + \frac{\beta}{\alpha}\frac{\|\widetilde{T}_l - T_l\|_F}{\|T_l\|_F}\right].$$

The last term may be bounded using the inductive hypothesis, but we must still bound $\|\widetilde{T}_l\|_F/\|T_l\|_F$. To economise on notation, let us construct the argument for $J_l$ rather than $T_l$:

$$\frac{\|\widetilde{J}_l\|_F}{\|J_l\|_F} = \frac{\|\widetilde{\Phi}'_L \widetilde{W}_L \widetilde{T}_l\|_F}{\|\Phi'_L W_L T_l\|_F} \leq \frac{\beta}{\alpha}\frac{\|\widetilde{W}_L \widetilde{T}_l\|_F}{\|W_L T_l\|_F} \leq \frac{\beta}{\alpha}\kappa^2\frac{\|\widetilde{W}_L\|_F\|\widetilde{T}_l\|_F}{\|W_L\|_F\|T_l\|_F} \leq \frac{\beta}{\alpha}\kappa^2\left(1 + \frac{\|\Delta W_L\|_F}{\|W_L\|_F}\right)\frac{\|\widetilde{T}_l\|_F}{\|T_l\|_F}.$$

By a simple induction, we then obtain:

$$\frac{\|\widetilde{J}_l\|_F}{\|J_l\|_F} \leq \prod_{k=l+1}^{L}\frac{\beta}{\alpha}\kappa^2\left(1 + \frac{\|\Delta W_k\|_F}{\|W_k\|_F}\right) \quad\Longrightarrow\quad \frac{\|\widetilde{T}_l\|_F}{\|T_l\|_F} \leq \prod_{k=l+1}^{L-1}\frac{\beta}{\alpha}\kappa^2\left(1 + \frac{\|\Delta W_k\|_F}{\|W_k\|_F}\right).$$

Since $\beta > \alpha$, we are free to relax the latter bound to:

$$\frac{\|\widetilde{T}_l\|_F}{\|T_l\|_F} \leq \left(\frac{\beta}{\alpha}\kappa^2\right)^{L-l-1}\prod_{k=l+1}^{L-1}\frac{\beta}{\alpha}\left(1 + \frac{\|\Delta W_k\|_F}{\|W_k\|_F}\right).$$

Similarly we are free to insert one additional factor of $\beta/\alpha$ into the first term of the bound on $\|\widetilde{J}_l - J_l\|_F / \|J_l\|_F$, to obtain:

$$\frac{\|\widetilde{J}_l - J_l\|_F}{\|J_l\|_F} \leq \frac{\beta}{\alpha} \kappa^2 \left[ \frac{\beta - \alpha}{\alpha} \left(1 + \frac{\|\Delta W_L\|_F}{\|W_L\|_F}\right) \frac{\|\widetilde{T}_l\|_F}{\|T_l\|_F} + \frac{\|\Delta W_L\|_F}{\|W_L\|_F} \frac{\|\widetilde{T}_l\|_F}{\|T_l\|_F} + \frac{\|\widetilde{T}_l - T_l\|_F}{\|T_l\|_F} \right]$$

We now substitute in the inductive hypothesis and the bound on $\|\widetilde{T}_l\|_F / \|T_l\|_F$ to obtain:

$$\frac{\|\widetilde{J}_l - J_l\|_F}{\|J_l\|_F}$$

$$\leq \left(\frac{\beta}{\alpha}\kappa^2\right)^{L-l} \left[ \left[ \frac{\beta - \alpha}{\alpha} \left(1 + \frac{\|\Delta W_L\|_F}{\|W_L\|_F}\right) + \frac{\|\Delta W_L\|_F}{\|W_L\|_F} + 1 \right] \prod_{k=l+1}^{L-1} \frac{\beta}{\alpha}\left(1 + \frac{\|\Delta W_k\|_F}{\|W_k\|_F}\right) - 1 \right]$$

$$= \left(\frac{\beta}{\alpha}\kappa^2\right)^{L-l} \left[ \left(1 + \frac{\beta - \alpha}{\alpha}\right) \left(1 + \frac{\|\Delta W_L\|_F}{\|W_L\|_F}\right) \prod_{k=l+1}^{L-1} \frac{\beta}{\alpha}\left(1 + \frac{\|\Delta W_k\|_F}{\|W_k\|_F}\right) - 1 \right]$$

$$= \left(\frac{\beta}{\alpha}\kappa^2\right)^{L-l} \left[ \prod_{k=l+1}^{L} \frac{\beta}{\alpha}\left(1 + \frac{\|\Delta W_k\|_F}{\|W_k\|_F}\right) - 1 \right],$$

which is what needed to be shown. $\square$

Table 2: Test set results. We quote loss for the classifiers, FID [4] for the GAN, and perplexity for the transformer—so lower is better. Training set results are given in Table 1.

| Benchmark | SGD $\eta$ | Fromage $\eta$ | Adam $\eta$ | SGD | Fromage | Adam |
|---|---|---|---|---|---|---|
| CIFAR-10 | 0.1 | 0.01 | 0.001 | $0.545 \pm 0.002$ | $\mathbf{0.31 \pm 0.02}$ | $0.76 \pm 0.02$ |
| ImageNet | 1.0 | 0.01 | 0.001 | $\mathbf{1.091 \pm 0.006}$ | $1.126 \pm 0.002$ | $1.184 \pm 0.009$ |
| GAN | 0.01 | 0.01 | 0.0001 | $34 \pm 2$ | $\mathbf{16 \pm 1}$ | $23.9 \pm 0.9$ |
| Transformer | 1.0 | 0.01 | 0.0001 | $\mathbf{169.6 \pm 0.6}$ | $171.1 \pm 0.3$ | $172.7 \pm 0.3$ |

## Appendix C  Experimental details

We provide the code used to run the experiments at `https://github.com/jxbz/fromage`. All experiments were run on a single NVIDIA Titan RTX GPU, except the ImageNet experiment which was distributed across 8 NVIDIA V100 GPUs.

We will now summarise the key details of the experimental setup.

### Figure 1: measuring the loss curvature

We train multilayer perceptrons of depth 2 and 16 with relu nonlinearity on the MNIST dataset [39]. Each $28\,\mathrm{px} \times 28\,\mathrm{px}$ image is flattened to a 784 dimensional vector. All weight matrices of the multilayer perceptron are of dimension $784 \times 784$, except the final output layer which is of dimension $784 \times 10$. To train the network, we minimise the softmax cross-entropy loss function on the network output. We use the Fromage optimiser with an initial learning rate of 0.01 and reduce the learning rate by a factor of 0.9 every epoch. A training minibatch size of 250 datapoints is used. We plot the training accuracy over the 10 epochs of training, and smooth these training accuracy curves over a window length of 5 iterations to improve their legibility.

During the 10 epochs of training we record 10 snapshots of the model weights. For the 2 layer network, we record snapshots more frequently during the first epoch since this is when most of the learning happens. The 16 layer network trains slower so we record snapshots once per epoch.

For each saved snapshot of the depth $L \in \{2, 16\}$ network, we now investigate properties of the loss surface and gradient for perturbations to that snapshot. Specifically, for every layer in the network we perturb the weights $W_l$ along the full batch gradient direction $g_l$. That is, for $\eta \in [0, 0.1]$ we record the loss $\mathcal{L}(\widetilde{W})$ and full batch gradient $\widetilde{g}_l$ for perturbed networks with parameters given by:

$$\widetilde{W}_l = W_l - \eta \cdot g_l \cdot \frac{\|W_l\|_F}{\|g_l\|_F} \qquad (l = 1, ..., L).$$

We plot the loss $\mathcal{L}(\widetilde{W})$ and relative change in gradient $\frac{\|\widetilde{g}_1 - g_1\|_F}{\|g_1\|_F}$ for the first network layer as a function of $\eta \in [0, 0.1]$.

### Figure 2 (right): stability of weight norms

With the same experimental setup as for class-conditional GAN training (see below), we run a lesion experiment on Fromage where we disable the $1/\sqrt{1 + \eta^2}$ prefactor. This makes Fromage equivalent to the LARS algorithm [6]. We plot the norms of all spectrally normalised layers in both the generator and discriminator during 100 epochs of training.

### Figure 3 (left): training multilayer perceptrons at large depth

With the same basic training setup as for Figure 1, this time we vary the depth of the multilayer perceptron and benchmark SGD, Adam and Fromage. The main difference to the Figure 1 setup is that this time we train for 100 epochs (to allow more time for learning to converge) and we decay the learning rate by factor 0.95 every epoch, so that the learning rate has reduced by 2 orders of magnitude after roughly 90 epochs of training. For each learning algorithm we run three initial learning rates at each depth: for SGD we try $\eta \in \{10^0, 10^{-1}, 10^{-2}\}$, for Fromage we try $\eta \in \{10^{-1}, 10^{-2}, 10^{-3}\}$

and for Adam we try $\eta \in \{10^{-2}, 10^{-3}, 10^{-4}\}$. These values were found to be well-suited to each algorithm in preliminary experiments. For Adam we set its $\beta_1$ and $\beta_2$ hyperparameters to the standard values of 0.9 and 0.999 suggested by Kingma and Ba [3]. For SGD we set the momentum value to 0.9, and a preliminary test suggested that this improved its performance versus switching off momentum.

**Figure 3 (right): learning rate tuning**

For each benchmark (full details below) we conduct a learning rate grid search. For each learning rate in $\{10^{-4}, 10^{-3}, 10^{-2}, 10^{-1}, 10^{0}\}$ we plot the error after a fixed number of epochs. No learning rate decay schedule is used here. In the CIFAR-10 classification experiment, we record training loss at epoch 50. In the GAN experiment, we record FID between the training set and generated distribution at epoch 100. In the transformer experiment, we record training perplexity at epoch 10.

**Class-conditional generative adversarial network training**

We train a class-conditional generative adversarial network with projection discriminator [33, 37] on the CIFAR-10 dataset [42]. Whilst our architecture is custom, it attempts to replicate the network design of Brock et al. [5]. We use the hinge loss for training, following Miyato and Koyama [37]. We train for 120 epochs at batch size 256, and divide the learning rate by 10 at epoch 100. We make one discriminator (D) step per generator (G) step. We use equal learning rates in G and D. For all algorithms we tune the initial learning rate on a logarithmic scale (over powers of 10).

To report accuracy, we use the FID score [4]. In essence, this score measures the distance between two sets of images by measuring the difference in the first and second moments of their representations at the penultimate layer of an `inception_v3` [43] classification network. It is intended to measure a notion of *semantic distance* between two sets of images. We report the FID score between the generated distribution and both the train and test set of CIFAR-10 to provide some indication of how well the learning generalises. We do not use post-processing techniques that have been found to improve FID scores such as the *truncation trick* [5] which adjusts the input distribution to the generator at test time with a tunable hyperparameter, or by reporting FID scores on an exponential moving average of the generator [44] which also introduces an extra tunable hyperparameter.

**ImageNet classification**

We train the `resnet50` network [40] on the ImageNet 2012 ILSVRC dataset [45] distributed over 8 V100 GPUs. We use a batch size of 128 images per GPU, meaning that the total batch size is 1024. The network is trained for a total of 90 epochs, with the learning rate decayed by a factor of 10 after every 30 epochs. A standard data augmentation scheme is used. The initial learning rate is tuned over the set $\{10^{-3}, 10^{-2}, 10^{-1}, 10^{0}, 10^{1}\}$ based on the best top-1 accuracy on the validation subset. The final results are reported on the test subset for three runs with different random initialization seeds. For the Adam optimiser, the $\beta_1$ and $\beta_2$ parameters are set to their default values of 0.9 and 0.999 as recommended by Kingma and Ba [3]. For SGD, the weight decay coefficient is set to $10^{-4}$ as recommended by He et al. [40].

**Wikitext-2 transformer**

We train a small transformer network [46] on the Wikitext-2 dataset [47]. The code is borrowed from the Pytorch examples repository at this https url. The network is trained for 20 epochs, with the learning rate decayed by 10 at epoch 10. Perplexity is recorded on both the training and test sets. We found that without regularisation, Fromage would heavily overfit the training set. We were able to correct this behaviour by bounding each layer's parameter norm to be smaller than its initial value.

**CIFAR-10 classification**

We train a `resnet18` network [40] on the CIFAR-10 dataset [42]. We train for 350 epochs and divide the learning rate by 10 at epochs 150 and 250. For data augmentation, a standard scheme is used involving random crops and horizontal flips. We report training and test loss. Again, we found that without regularisation, Fromage would heavily overfit the training set. And again, we were able to correct this behaviour by bounding each layer's parameter norm to be smaller than its initial value.