[Reviews · NeurIPS 2020]

Review 1

Summary and Contributions: The paper starts by mentioning the difficulty in choosing an appropriate learning rate and motivates the need for eliminating grid search in tuning the learning rate. The authors proceed to derive bounds on the relative distance in both the functional value and the gradients between the two structured multilayer perceptrons in terms of the perturbations of the parameters in each layer. From this, the authors model the gradient breakdown of the network loss function, termed as deep relative trust, and derive a descent lemma for neural networks. The proposed optimization algorithm, called Fromage, claims to ease the choice of an appropriate learning rate without the need to tune it through a grid search. In addition, the proposed Fromage may stabilize the network by avoiding the exploding gradient problem.

Strengths: The problem studied is interesting and worthwhile. The proposed algorithm may require less effort in deploying a neural network because of the ease in choosing the learning rate. The authors discuss the possible reason for gradient breakdown through perturbation analysis which led to the proposed learning rule. The claims and experimental validations are fairly sound but not thorough.

Weaknesses: The proposed learning rule is somewhat similar to Layer-wise Adaptive Rate Scaling (LARS) by You et al. except for the additional weight decay. The authors should consider comparing the proposed approach with LARS in Table 1 of page 8. For the experimental results reported in Table 1, the authors implemented various networks using the same learning rate for Formage. The authors showed that the learning rate for L<50 could be 0.01. However, the networks in Table 1 are very different and hence, different learning rates could be chosen based on the structure of the network using lemma 2. The authors may consider choosing a learning rate specific to the network to show the results of table 1 to validate lemma 2. Other possible issues with Fromage include the test accuracy for classification networks provided in the supplementary material. However, for GANs, an important question is the quality of the generated images, which is not discussed. Lemma 2 implies that the learning rate should be small for larger networks. How does that affect the convergence? What if the learning rate is further reduced? In Figure 1- page 5, the gradient break down is shown for a 16 layer perceptron using Fromage. Would not it be better to show the gradient break down using SGD? The authors should also consider discussing a case where gradient breakdown does not occur in the standard SGD to see the difference between SGD and Formage in that setting. I am leaving my score as is since the authors' responses were not strong enough.

Correctness: The claims appear to be correct but did not check the proofs carefully.

Clarity: The paper is well written and the mathematical results are explained at an intuitive level.

Relation to Prior Work: Prior work is cited appropriately as far as I can tell. However the authors discuss the common tricks to avoid the vanishing gradient or exploding gradient problem, mentioning only the issues with the careful choice of weight initialization [8]. The authors may consider discussing the limitations of other techniques, like dividing out the gradient scale [3] or gradient clipping [9] and how these techniques differ from the proposed one.

Reproducibility: Yes

Additional Feedback: Several suggestions under Weaknesses were made which can improve the paper.


Review 2

Summary and Contributions: This paper proposes the Fromage algorithm to optimize deep neural networks. Fromage is motivated by replacing the Euclidean geometry induced by the isotropic quadratic penalty term underlying the vanilla gradient descent into a distance that is tailored to multilayer neural networks. The distance is developed by analyzing the relative error. The algorithms turns out to be a rescaled version of layerwise adaptive rate scaling (LARS). Empirical results show promise of the proposed algorithm in terms of the performance and amount of tuning.

Strengths: (1) The algorithm is well motivated. (2) The insights from the relative error analysis is helpful in algorithmic design for training deep neutral networks. (3) The paper is well-written.

Weaknesses: I found a few parts confusing but it may be due to my misunderstanding. So please correct me if I am wrong. (1) Suppose Algorithm 1 converges to a stationary point (or even local/global minimum) x*, then x* = x* / sqrt(1 + \eta^2), implying that x* = 0. In other words, Algorithm 1 can converge only if 0 is a stationary point, in which case 0 is the only point that the algorithm can converge to. This seems to be strange to me. Do the authors observe an oscillating or non-converging behavior of the iterates in the examples? (2) In You et al. (2017), the LARS algorithm (without acceleration) can be written as w = w - \gamma \ell (\beta w + ||w|| (\nabla w) / (1 + \beta)||\nabla w||) = (1 - \gamma\ell\beta)w - (\gamma\ell) / (1 + \beta) ||w||(\nabla w) / ||\nabla w||. It is easy to find \gamma, \ell and \beta to recover the update in Fromage. (3) The modelling assumption 1 is confusing. Compared to Theorem 1, it neglects a huge constant C = (\beta / \alpha \kappa^2)^L that scales exponential to L and replaces it into 1. Whereas I understand that the goal of Theorem 1 is to illustrate the intuition, Lemma 2 is crucial in justifying the choice of 0.01 for \eta. If the constant C is added back, \eta need to be below (1 + C^{-1}\cos\theta)^{1/L} - 1 \approx \cos\theta / (LC). This decays exponentially with the number of layers, which appears to conflict with the goal of this paper. (4) line 146 - 147 "Although relu is technically not covered by Theorem 1, we may model the fact that it transmits half its input domain by setting". What does this mean? (5) When I read the first paragraph, I thought the paper is going to modify the quadratic term and apply the steepest descent on the modified one. But Fromage does not seem to work that way. Could you elaborate the connection between the algorithm and the motivation in Section 1? ------------------------ Post-rebuttal update ----------------------- I thank the authors for their responses. The rebuttal addressed most of my concerns so I raised my score from 5 to 6. I still think ignoring the exponentially growing constant significantly undermines the theory, which is the main contribution of this paper. It would strengthen the paper a lot if the authors can show how BatchNorm or other techniques reduces this constant theoretically, even under strong assumptions.

Correctness: See above.

Clarity: Yes

Relation to Prior Work: Yes

Reproducibility: Yes

Additional Feedback:


Review 3

Summary and Contributions: To study the gradient breakdown of nonlinear compositional functions, this paper proposes with a distance function between neural networks called deep relative trust by analyzing the perturbation of neural network functions and Jacobians. Based on the deep relative trust, the paper yields a descent lemma for neural networks. Then based on this descent lemma, a new optimizing algorithm Fromage is proposed. Given that it has only one hyperparameter, Fromage does not require the tuning of the learning rate for training neural networks. Finally, this paper deploys several specific experiments using GAN, natural language transformers and image classifiers to prove that generally Fromage performs better than some common optimizers such as SGD and Adam.

Strengths: 1. The logic of this paper is rigorous and clear. After the introduction, some basics of deep learning (e.g., multilayer perceptron) are introduced, including heuristics of algorithm design and drawbacks of common optimization theory. Then by proposing the deep relative trust, the definitions of relative functional distance and gradient distance between neural networks are provided to explore how to avoid tuning the learning rate for a perturbation and measure the empirical loss effect. Afterward, the new descent rule under the deep relative trust is studied to account for the neural network structure. Finally, a new learning algorithm Fromage is proposed based on previous theory results, and extensive experiments are deployed to validate its advantages and effectiveness. 2. The theory is sound and complete. In the theoretical part, a definition of Multilayer Perceptron (MLP) and a proposition of Jacobian of MLP are provided. To introduce the distance between neural networks, Theorem 1 is proposed to explain that the relative change of MLP's function and gradient is controlled by a product over relative perturbations to each layer. And the quasi-exponential at large depth L is concluded, as a growth property of trust region. Afterward, the feasibility of the assumptions of Theorem 1 is explored. Then it yields the descent lemma, Modelling assumption 1 (deep relative trust) and Lemma 2 to provide the theory support for the following algorithm. Finally, the new optimizer Fromage is proposed and compared with current common optimizers. 3. The Fromage optimizing algorithm is practical in real-world applications as it requires only one hyperparameter tuning compared with SGD, which can save training time and resources, and provide convenience to train an advanced neural network.

Weaknesses: 1. The proposed Fromage algorithm is similar to LARS that has already been proposed by You et al. (2017). As commented by the author, the only difference between LARS and Fromage is that the latter provides a prefactor to avoid the numerical overflow. In fact, when eta is relatively small, the main update of Fromage can be understood as LARS with a weight decay.

Correctness: Yes, I think the claims, theoretical methods, and empirical methodology are all correct.

Clarity: Yes, I think the paper is well written.

Relation to Prior Work: Yes. For example, the authors clearly stated that their proposed Fromage differs to LARS [You et al., 2017] by adding a prefactor to this algorithm which avoid the numerical overflow.

Reproducibility: Yes

Additional Feedback: In fig1, the authors examines the gradient of a 16-layer MLP to show that deep networks violates the Euclidean smoothness geometry. This is an interesting observation. However, very deep MLP is rarely used in practice, as the training can be unstable, this matches fig 1 that shows that such a deep MLP has a bad landscape. On the other hand, the authors may want to examine other models such as 3-layer MLP and Resnet-16 and see whether these trainable (deep) models violates the Euclidean smoothness or not. It is possible that these trainable networks satisfy the smoothness condition around the global min, as justified by many empirical and theoretical works on NN landscape.

[Author Response · NeurIPS 2020]

**All reviewers & AC** Thanks to all the reviewers for their valuable feedback. We would like to begin by clarifying our main contributions:

(1) We developed a **new framework for first order optimisation** (Lemma 1). Unlike existing first order optimisation frameworks—such as *mirror descent* and *steepest descent*— where it is not clear which distance metric to use, our framework makes a strong case that the correct distance metric measures the **relative breakdown in gradient**.

(2) We bounded the **relative breakdown in gradient for multilayer perceptrons** (Theorem 1). This provides a principled way to derive deep learning algorithms. Moreover, we believe this application of **perturbation theory to understand deep learning** is highly original and could inspire much followup work.

(3) We derived the Fromage algorithm, which is **admittedly very similar to LARS**. Besides the fact that LARS is susceptible to numerical overflow (Figure 2), Fromage and LARS have very similar performance. Whilst we acknowledge this, we believe that it **does not detract from our main contributions**.

**Reviewer 1** Thank you for your valuable comments.

(1) The inset figure shows the performance of Fromage for training MLPs of varying depth using three learning rates. The figure shows that a smaller learning rate is needed to train deeper MLPs, as suggested by Lemma 2. The fact that Fromage with $\eta = 0.01$ trained the off-the-shelf networks in Table 1 was a surprising empirical observation.

(2) In the GAN literature, FID score is a standard proxy for sample quality.

(3) Fromage is used in Figure 1 because we could not get SGD to train such deep multilayer perceptrons.

(4) On *dividing out the gradient scale*—this approach (taken by Adam) requires more learning rate tuning than Fromage. *Gradient clipping*—this approach requires tuning both the learning rate *and* the clipping threshold.

**Reviewer 2** These are great questions. We shall address each in turn.

(1) For fixed learning rate $\eta$, the iterates do "jitter" around a minimum. This is because Fromage always induces a relative change of $\eta$ in the weights. Convergence is attained by decaying $\eta$ when the loss plateaus. [If you are wondering how this fits with our optimisation theory, let us again inspect Equation 2. Close to a minimum of the loss function $\mathcal{L}$, the term $\partial\mathcal{L}/\partial f$ will become the main cause of relative change in the gradient $\nabla_{W_l}\mathcal{L}$, but our deep relative trust model neglects this term $\partial\mathcal{L}/\partial f$.]

(2) Fromage and LARS are similar, but **LARS is empirically motivated and lacks theoretical analysis.**

(3) The large constant factor in Theorem 1 is a result of assuming that the *worst case* happens at every network layer. In practice this is too pessimistic, so we neglect the constant factor in modelling assumption 1. It is also plausible that in modern network architectures, conditioning techniques like BatchNorm and skip connections improve the constant factor. This is a good line of inquiry for further research.

(4) RELU takes an input in $\mathbb{R}$ and projects it on to $\mathbb{R}^+$—thus RELU "transmits half its input domain". For a vector of inputs $x$, a reasonable model is $\|\mathrm{relu}(x)\|_2 = \|x\|_2/2$. This model may be more realistic when BatchNorm is used, since BatchNorm centers the average input to RELU.

(5) To compare our proposed optimisation framework to steepest descent, note that Lemma 1 can be rewritten as:
$$\mathcal{L}(W + \Delta W) \leq \mathcal{L}(W) + g(W)^T \Delta W + \max_{t\in[0,1]} \|g(W + t\Delta W) - g(W)\|_2 \cdot \|\Delta W\|_2.$$

From this perspective, our framework is analogous to steepest descent with distance: $\max_{t\in[0,1]} \|g(W + t\Delta W) - g(W)\|_2 \cdot \|\Delta W\|_2$. The advantage over steepest descent is that our scheme explicitly tells you how to build your distance metric—you must measure or bound the breakdown in gradient $\max_{t\in[0,1]} \|g(W + t\Delta W) - g(W)\|_2$.

**Reviewer 3** Thank you for your review.

(1) We present the analogue of Figure 1 for a 16-layer residual network in the inset figure. As can be seen, this model does violate the Euclidean smoothness assumption—the gradient and loss breakdown are both quasi-exponential. But the breakdown is less severe than for the 16-layer network without skip connections—suggesting that skip connections do improve the loss landscape.



[Meta-Review · NeurIPS 2020]

Three reviewers indicate "weak accept". They acknowledge the theoretical deviation and the proposed algorithm. One reviewer is still concerned that "ignoring the exponentially growing constant significantly undermines the theory" which I agree. I read the paper and find a few other issues: --The fact that the fixed point (adding prefactor) is not stationary point is not satisfactory. I think there is a way to resolve this, though cannot think of it now. --R1 asked about why the theory requires layer-dependent eta, but simulation shows 0.01 works for different tasks. The rebuttal adds experiments show deeper MLP requires smaller eta, which somewhat matches theory, which is good and should be added to Figure 3. Figure-3-right shows 0.01 works for three tasks; I thought this is due to they use somewhat similar layers, but the authors seem to think it is "pleasant surprise". If the three tasks (classification, GAN, transformer) use similar layers, please add "the # of layers are ..." into the caption and explain that it matches the theory; if they do not use similar layers, then Figure-3-right is somewhat misleading and more discussions on the caption are needed (e.g. add "this is unexpected and not predicted by the theory") since otherwise readers may treat Fig-3-right as validating "less hyper parameter tuning" in the abstract (or got confused if they saw "one tuning hyperparameter" but now found it does not require tuning). --It is not clear what the authors mean by "only one hyperparamter". If one uses SGD, then still only one hyperparameter. It seems the authors are comparing with Adam or other more complicated methods which require tuning. If the authors imply SGD requires tuning "decreasing xx times at xx epochs", then Fromage also did this. --quasi-exponential growth is expected, even without the derivation of this paper. It is a well-known fact due to layer-accumulation. The figure still shows a local quadratic behavior, and that is what people typically refer to. --In Thm 1, an unsaid assumption is that the weight matrix should be square or fat (or W' is fat; only one shape is possible) to ensure "condition number kappa is finite". I suggest to define "condition number of W" more rigorously in Thm 1 and state explicitly the restriction of weight matrices. --Show loss values but not accuracy is rather unconventional. The training acc and/or test acc need to be added. --Why checking conditional GAN, not standard GAN experiments like in SN-GAN paper? That is more standard for a generic paper. The architecture and hyperparamters are important factors for GAN results. E.g. using the same lr for G and D is known to be sub-optimal (check TTUR for GANs). Please modify the paper accordingly. Despite the issues, this paper directly relates a theoretical analysis to a new algorithm, which is interesting. In addition, the simulation results are quite good (though limited). For these reasons, I recommend accept.